# Accelerating Inference of Discrete Autoregressive Normalizing Flows by Selective Jacobi Decoding

**Jiaru Zhang**[*]                                          *jiaru@purdue.edu*
*Institute of Physical Artificial Intelligence*
*Purdue University*

**Juanwu Lu**                                          *juanwu@purdue.edu*
*College of Engineering*
*Purdue University*

**Xiaoyu Wu**                                          *xw105@rice.edu*
*Rice University*

**Ziran Wang**[†]                                          *ziran@purdue.edu*
*College of Engineering*
*Purdue University*

**Ruqi Zhang**[†]                                          *ruqiz@purdue.edu*
*Department of Computer Science*
*Purdue University*

**Reviewed on OpenReview:** *https://openreview.net/forum?id=xYATz9HpE7*

## Abstract

Discrete normalizing flows are promising generative models with advantages such as analytical log-likelihood computation and end-to-end training. However, the architectural constraints to ensure invertibility and tractable Jacobian computation limit their expressive power and practical usability. Recent advancements utilize autoregressive modeling, significantly enhancing expressive power and generation quality. Nevertheless, such sequential modeling inherently restricts parallel computation during inference, leading to slow generation that impedes practical deployment. In this paper, we first identify that strict sequential dependency in inference is unnecessary to generate high-quality samples. We observe that sub-variables in sequential modeling can also be approximated without strictly conditioning on all preceding sub-variables. Moreover, the models tend to exhibit low dependency redundancy in the initial layer and higher redundancy in subsequent layers. Leveraging these observations, we propose to selectively use Jacobi decoding strategy that accelerates its autoregressive inference through parallel iterative optimization. Theoretical analyses demonstrate the method's superlinear convergence rate and guarantee that the number of iterations required is no greater than the original sequential approach. Empirical evaluations across multiple datasets validate the generality and effectiveness of our acceleration technique, achieving up to 4.7 times faster inference on modern normalizing flow models while preserving generation quality.

---

[*]Correspondence to: Jiaru Zhang.    [†]Equal advising.

# 1 Introduction

Discrete normalizing flow models (*a.k.a. normalizing flows*) have emerged as promising generative models by modeling the transformation between a data variable and a latent Gaussian variable through a series of mappings, where the mapping functions are constructed as invertible functions parameterized by neural networks. This approach allows for end-to-end training with a single loss function, ensuring consistency between encoding and decoding. Moreover, the model's structure facilitates the computation of the analytical log-likelihood. These benefits have attracted significant interest from the research community, positioning discrete normalizing flow models as a compelling tool for generative modeling (Dinh et al., 2015; 2017; Ho et al., 2019; Kingma & Dhariwal, 2018; Papamakarios et al., 2021).

Although discrete normalizing flow models are theoretically rigorous, they often exhibit limited generation capabilities in practice due to unavoidable architectural constraints imposed by the requirement for invertible mappings and the need to compute Jacobian matrices. For example, a classical construction of the invertible function, i.e., the affine coupling layer, is to split the input into two parts, and the output is obtained by the concatenation of those processed separately by trainable networks (Dinh et al., 2015; 2017; Kingma & Dhariwal, 2018). This formulation ensures the analytical solution of both the invertible function and the Jacobian matrix, but it also yields relatively limited expressive power and less compatible network architectures, which affect the quality of generated samples and practical applications. To address this, the recently proposed TarFlow (Zhai et al., 2025) model employs a block autoregressive architecture inspired by autoregressive normalizing flows (Kingma et al., 2016; Papamakarios et al., 2017). It splits the input into a longer sequence rather than a couple and constructs the invertible function using masked autoregressive transformations. This autoregressive modeling can be naturally incorporated into the causal vision transformer architecture, which provides a powerful representation, enabling TarFlow to achieve state-of-the-art performance in both density estimation and image synthesis.

However, while such autoregressive sequential modeling endows the model with powerful generative capabilities, it also incurs high parallel computational complexity during inference. Concretely, the inverse function of the sequential construction is an autoregressive inference process, meaning that each new sub-variable is generated sequentially, relying on all of the previously generated sub-variables. It limits parallel computation, thereby slowing generation speeds and restricting practical applications, as also noted in previous work (Zhai et al., 2025).

In this work, we first identify that strict sequential conditioning contains significant, layer-varying redundant dependencies during inference. Specifically, we observe that subsequent sub-variables can still be approximated without the nearest preceding sub-variables. Moreover, we find that the amount of redundancy varies substantially across layers, with the later layers exhibiting substantially more redundancy than the first. Motivated by these observations, we propose to selectively use the Jacobi decoding approach to accelerate the inference of discrete autoregressive normalizing flow models. Our approach leverages parallel iterative optimization to achieve fast convergence to high-fidelity samples, without requiring additional training or modifications to the original model architecture, and is backed by theoretical convergence guarantees of superlinear speed and finite convergence. Experimental results on diverse datasets, including CIFAR-10, CIFAR-100, and AFHQ (Choi et al., 2020), verify the generality and effectiveness of the proposed acceleration approach. The code is available at `https://github.com/lan-qing/SJD`. In summary, our contributions include:

**C1** We discover that strict dependency on the original sequential inference of discrete autoregressive normalizing flows contains redundancy, and the quantity of the redundancy varies across different layers, based on our theoretical analysis and empirical observation.

**C2** We introduce an approach to accelerate inference in discrete autoregressive normalizing flows by selectively applying Jacobi decoding in inference. Theoretical analysis demonstrates superlinear convergence and a finite convergence guarantee.

**C3** We conduct comprehensive experiments to validate the effectiveness of our approach, showing up to 4.7 times speed improvements with little impact on generation quality across diverse datasets, including CIFAR-10, CIFAR-100, and AFHQ.

## 2   Related Work

**Inference Acceleration of Generative Models.**   Inference acceleration is critical for the practical application of generative models, and researchers have proposed various effective strategies for different generative models. For Diffusion Models (DMs), techniques such as improved numerical solvers (Dockhorn et al., 2022; Lu et al., 2022; Song et al., 2021a), knowledge distillation (Salimans & Ho, 2022), and consistency modeling (Song et al., 2023) have significantly reduced sampling times. For Variational Autoencoders (VAEs) and Generative Adversarial Networks (GANs), methods like network pruning (Kumar et al., 2023; Saxena et al., 2024), quantization (Andreev & Fritzler, 2022), and knowledge distillation (Aguinaldo et al., 2019; Yeo et al., 2024) are widely adopted to enhance inference efficiency. These approaches, however, typically leverage specific architectural properties or training paradigms inherent to these models. Consequently, they are generally not directly transferable to inference acceleration of autoregressive normalizing flows. To the best of our knowledge, this work is the first to explore the acceleration of inference for normalizing flow models.

**Jacobi Decoding.**   Inspired by the research on non-linear equation solutions (Ortega & Rheinboldt, 2000), Jacobi decoding has emerged as a promising approach to accelerating neural network inference. It aims to break the sequential dependency by reformulating generation as an iterative process of solving a system of equations, often framed as a fixed-point problem. Song et al. (2021b) first developed a theoretical framework for interpreting feedforward computation as the solution of nonlinear equations and revealed the significant potential of Jacobi decoding for networks such as RNNs and DenseNets. Santilli et al. (2023) further confirms the effectiveness of Jacobi decoding on the language generation task. It was further improved with additional fine-tuning to keep the consistency of decoded tokens (Kou et al., 2024). In image generation, Teng et al. (2025) explored the inference acceleration by the combination of Jacobi decoding with a probabilistic criterion for token acceptance on autoregressive text-to-image generation models. Although previously applied to language and image generation, these methods commonly encounter issues such as quality degradation (Teng et al., 2025) and limited acceleration in discrete token spaces (Santilli et al., 2023).

## 3   Methodology

This section introduces the method that selectively applies Jacobi decoding to accelerate discrete autoregressive normalizing flows. It improves inference efficiency by breaking dependencies among sub-variables. We begin with an introduction to normalizing flows (Section 3.1). Motivated by observations of sequential redundancy and its depthwise heterogeneity (Section 3.2), we propose parallel inference using Jacobi iterations (Section 3.3). We show that this method converges superlinearly and in finite time (Section 3.4). In addition, we propose a strategy that applies parallel Jacobi decoding only to higher-redundancy layers, thereby further improving efficiency (Section 3.5). To promote readability, we also present a table of notations in Appendix A.

### 3.1   Discrete Autoregressive Normalizing Flow

Normalizing flow is a family of generative models that explicitly learn a differential bijection $\boldsymbol{x} = f(\boldsymbol{z})$ between a latent random variable $\boldsymbol{z}$ and data $\boldsymbol{x}$ by leveraging the change of variable law (Dinh et al., 2015; Rezende & Mohamed, 2015). The optimal transformation is given by maximizing the log-likelihood of observed data

$$f^* \leftarrow \arg\max_f \log p(\boldsymbol{x}) = \log p(\boldsymbol{z}) - \log \det \mathbf{J}_f, \tag{1}$$

where $\mathbf{J}_f$ is the Jacobian matrix of $f$. To facilitate capturing of more complex data distribution $\boldsymbol{x} \sim p_{\text{data}}(\boldsymbol{x})$, a discrete normalizing flow learns not a single but a cascade of $K$ intermediate bijections such that

$$\boldsymbol{x} = f(\boldsymbol{z}_K) \triangleq (f_0 \circ f_1 \circ \ldots \circ f_{K-1})(\boldsymbol{z}_K). \tag{2}$$

The optimal set of bijections is then given by

$$f^* \leftarrow \arg\max_{f_0,\ldots,f_{K-1}} \log p(\boldsymbol{z}_K) - \sum_{k=0}^{K-1} \log \det \mathbf{J}_{f_k}. \tag{3}$$

However, properly constructing the bijections $f_k$ for high-dimensional variables is tricky. A commonly adopted method, coupling-based normalizing flow (Dinh et al., 2015), splits the high-dimensional $z_k$ into a pair of sub-variables $z_k = [z_{k,1}, z_{k,2}]^{\mathsf{T}}$, and has inspired consecutive works that combine neural networks for modeling by constructing building blocks such as the real-valued non-volume preserving (RealNVP) (Dinh et al., 2017), invertible $1 \times 1$ convolution (Kingma & Dhariwal, 2018), and self-attention layers (Ho et al., 2019).

More recent studies propose an autoregressive paradigm for discrete normalizing flow, which extends the above formulation by arbitrarily splitting random variables into a sequence of $L$ sub-variables (Kingma et al., 2016; Papamakarios et al., 2017). Then, the bijections $f_k$ can be defined by

$$
\begin{aligned}
z_{k+1} = f_k(z_k) &= f_k \left( \begin{bmatrix} z_{k,1} & z_{k,2} & \dots & z_{k,L} \end{bmatrix}^{\mathsf{T}} \right) \\
&\triangleq \begin{bmatrix} z_{k,1} \\ (z_{k,2} - g_k(z_{k,<2})) \odot \exp(s_k(z_{k,<2})) \\ \vdots \\ (z_{k,L} - g_k(z_{k,<L})) \odot \exp(s_k(z_{k,<L})) \end{bmatrix},
\end{aligned}
\tag{4}
$$

where $\odot$ denotes Hadamard product, $s_k(\cdot)$ and $g_k(\cdot)$ are functions to learn during training. Note that a permutation (e.g., reversing) of $z_{k+1}$ is also applied after each transformation to ensure all locations in the sequence can be transformed through the entire flow, while we omit it for simplicity. The bijection above naturally yields an inverse:

$$
\begin{aligned}
z_k = f_k^{-1}(z_{k+1}) &= f_k^{-1} \left( \begin{bmatrix} z_{k+1,1} & \dots & z_{k+1,L} \end{bmatrix}^{\mathsf{T}} \right) \\
&\triangleq \begin{bmatrix} z_{k+1,1} \\ z_{k+1,2} \odot \exp(-s_k(z_{k,<2})) + g_k(z_{k,<2}) \\ \vdots \\ z_{k+1,L} \odot \exp(-s_k(z_{k,<L})) + g_k(z_{k,<L}) \end{bmatrix}.
\end{aligned}
\tag{5}
$$

Discrete autoregressive normalizing flows have shown superiority in density estimation and image generation (Zhai et al., 2025) by integrating them into architectures such as the causal vision transformer. However, a critical issue arises during inference, which involves computing the inverse transformation. As suggested by equation 5, generating $z_{k,l}$ depends on all previous generations $z_{k,<l}$, which prohibits parallel computation and leads to slow sampling that restricts practical use cases. To address this issue, we leverage two fundamental observations on redundancy, presented in the next section.

### 3.2 Redundancy

**Sequential Redundancy.** As shown in equation 5, the generation process defined by the inverse transformation reveals a sequential dependency between each sub-variable and all preceding sub-variables. Consequently, the original inference procedure enforces a strictly sequential generation paradigm, significantly limiting generation speed. We hypothesize that this strict dependence is redundant, particularly for data such as images, which exhibit inherent spatial locality and continuity. Therefore, an element $z_{k,l}$ might be reasonably inferred even without precise, up-to-the-moment information from all its predecessors.

To validate our hypothesis, we conducted experiments using a straightforward transformation during inference, where the transformation step for element $z_{k,l}$ is modified to ignore information from the $o$-nearest preceding elements in the sequence by

$$
z_{k,l} = z_{k+1,l} \odot \exp(-s_k(z_{k,<(l-o)})) + g_k(z_{k,<(l-o)}),
\tag{6}
$$

for $l > 1$, where $z_{k,<(l-o)}$ is obtained by masking out the nearest $o$ preceding sub-variables in the attention operation. The experimental results shown in Figure 2 indicate that while the image quality diminishes as more of these nearest preceding sub-variables are removed, the model is still capable of generating meaningful images. It supports the claim that strict sequential dependencies contain potentially exploitable redundancy.

**Depthwise Heterogeneity of Redundancy.** We further investigate whether the degree of sequential redundancy varies across different layers during the generation process ($z_K \rightarrow \cdots \rightarrow x$). Theoretically, we

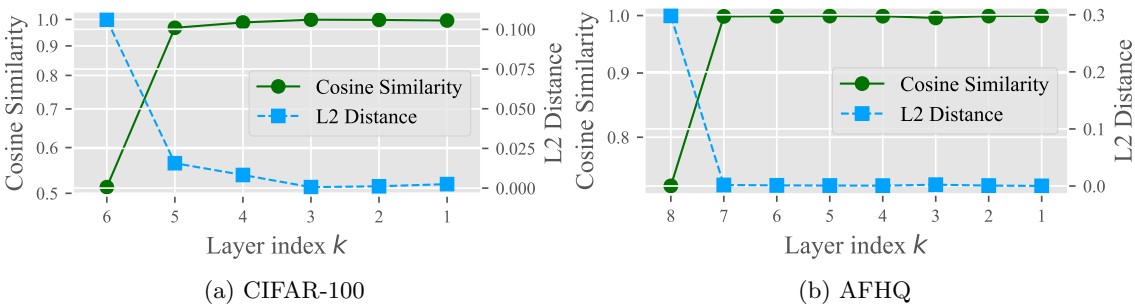

(a) CIFAR-100                    (b) AFHQ

Figure 1: Cosine similarities and L2 distances between layer outputs from standard inference and inference with $o = 5$ nearest preceding dependencies masked. Results of $o = 1$ and $o = 2$ are available in Figure A1.

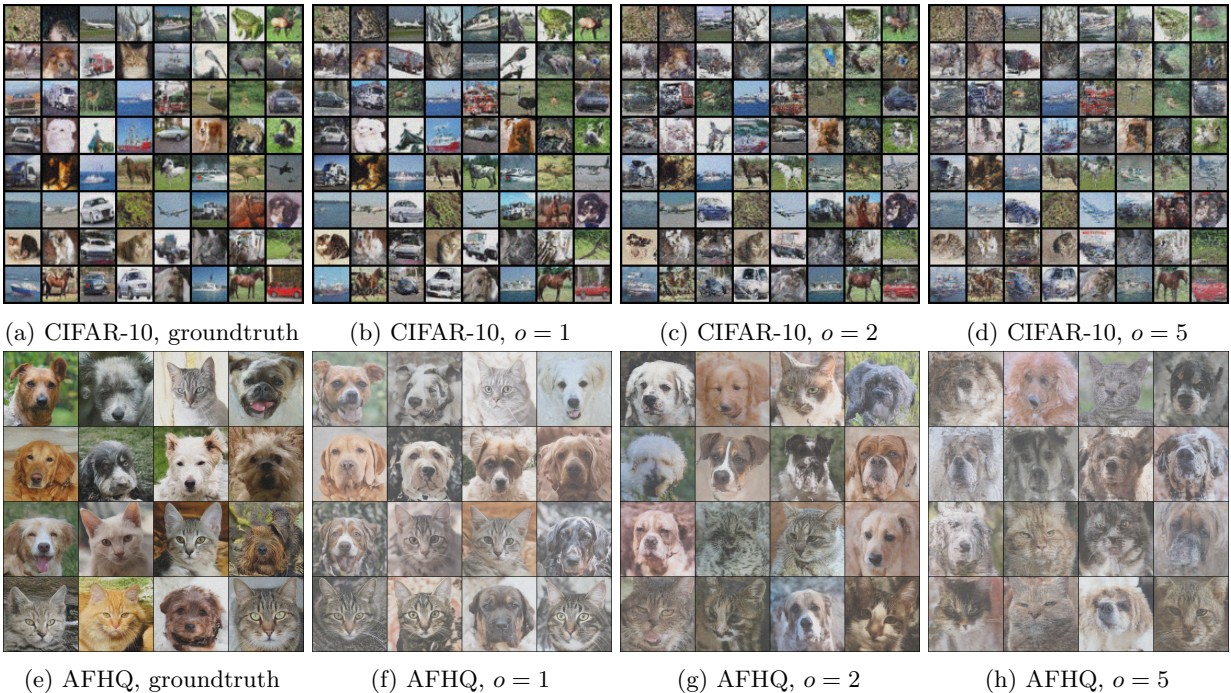

(a) CIFAR-10, groundtruth    (b) CIFAR-10, $o = 1$    (c) CIFAR-10, $o = 2$    (d) CIFAR-10, $o = 5$

(e) AFHQ, groundtruth    (f) AFHQ, $o = 1$    (g) AFHQ, $o = 2$    (h) AFHQ, $o = 5$

Figure 2: Generations where dependency on the nearest $o$ sub-variables is masked. It can still generate meaningful images, indicating the potential feasibility of acceleration with parallel computing.

expect heterogeneity: random variables $z_K$ from the first layer, which performs structure initiation from a Gaussian noise, tend to exhibit high dependency on preceding sub-variables, as the pure noise input contains theoretically no information and hence relies more on context given by preceding generations $z_{K,<l}$. Conversely, subsequent transformations refine informative outputs from the previous transformation $z_{k+1,l}$, leading to weak sequential dependency regarding preceding generations $z_{k,<l}$.

To verify this, we measure the cosine similarity and L2 distance deviation between the standard inference outputs $z_k$ and those generated while masking the $o$ nearest dependencies as shown in equation 6. As shown in Figure 1, the deviation is significantly larger for the first layer compared to subsequent ones. This result empirically confirms the low redundancy in the first layer, which is consistent with our expectation. The minimal deviation in subsequent layers aligns with their refinement role, which leverages informative inputs and the existing contextual structure. This observation motivates exploring layer-specific optimizations for the generation process.

### 3.3 Parallel Inference by Jacobi Iteration

---

**Algorithm 1:** Jacobi decoding for $f_k$

---

**Input:** Sequence $\boldsymbol{z}_{k+1}$, functions $s_k(\cdot)$ and $g_k(\cdot)$, stopping threshold $\tau$
**Output:** Sequence $\boldsymbol{z}_k$
Initialize $\boldsymbol{z}_k^0 = \boldsymbol{0}$, $t = 0$
**while** true **do**
    $t \leftarrow t + 1$, $\boldsymbol{z}_{k,1}^t \leftarrow \boldsymbol{z}_{k+1,1}$
    **for** $l = 2, \ldots L$
    **do in parallel**    $\boldsymbol{z}_{k,l}^t \leftarrow \boldsymbol{z}_{k+1,l} \odot \exp(-s_k(\boldsymbol{z}_{k,<l}^{t-1})) + g_k(\boldsymbol{z}_{k,<l}^{t-1})$
    **end for**
    **if** $\|\boldsymbol{z}_k^t - \boldsymbol{z}_k^{t-1}\|_\infty < \tau$
      **break**
    **end if**
**end while**
$\boldsymbol{z}_k = \boldsymbol{z}_k^t$

---

Our empirical observations of sequential redundancy revealed a key property of inference in discrete autoregressive normalizing flow: *the generation of subsequent element $\boldsymbol{z}_{k,l}$ exhibits a degree of robustness to inaccuracies in the preceding elements $\boldsymbol{z}_{k,<l}$.* This observation suggests that the strict, fully converged sequential dependency enforced by the standard inference procedure might contain redundancies and is not strictly necessary at every step for generation quality. This finding motivates exploring parallel computation strategies that can exploit this robustness.

To enable parallelization, we first re-examine the inference task. As established, generating the target sequence $\boldsymbol{z}_k$ from the input $\boldsymbol{z}_{k+1}$ via the inverse transformation defined in equation 5 fundamentally requires finding the unique solution $\boldsymbol{z}_k$ that satisfies the entire set of autoregressive conditional dependencies. It can be formally viewed as solving a system of $L$ non-linear equations $\mathcal{F}_l$, implicitly defined for $l = 1, \ldots, L$ as

$$\mathcal{F}_l(\boldsymbol{z}_{k,l}, \boldsymbol{z}_{k,<l}, \boldsymbol{z}_{k+1,l}) = 0, \tag{7}$$

where $\mathcal{F}_l = 0$ represents the condition imposed by the $k$-th step of the inverse transform, given the known input $\boldsymbol{z}_{k+1,l}$. The standard sequential inference method implicitly solves this system using a Gauss-Seidel-like approach, where the computation of $\boldsymbol{z}_{k,l}$ relies on the immediately preceding, fully computed values $\boldsymbol{z}_{k,1}, \ldots, \boldsymbol{z}_{k,l-1}$.

Leveraging the potential for parallelism indicated by our observations in Section 3.2, we propose employing the Jacobi decoding method to solve the system defined by equation 7. Instead of sequential updates, it performs iterative, parallel updates. Starting from an initial estimate $\boldsymbol{z}_k^0$, each iteration $t + 1$ computes a new estimate $\boldsymbol{z}_k^{t+1}$ where every element $\boldsymbol{z}_{k,l}^{t+1}$ is calculated based *only* on the elements from the previous iterate $\boldsymbol{z}_k^t$ and the output from previous layer $\boldsymbol{z}_{k+1}$:

$$\text{For } l = 1, \ldots, L, \text{solve for } \boldsymbol{z}_{k,l}^{t+1} \text{ from:} \\ \mathcal{F}_l(\boldsymbol{z}_{k,l}^{t+1}, \boldsymbol{z}_{k,<l}^t, \boldsymbol{z}_{k+1,l}) = 0. \tag{8}$$

Because the calculation of each $\boldsymbol{z}_{k,l}^{t+1}$ within an iteration only depends on values from the completed previous iteration $\boldsymbol{z}_k^t$, all $L$ updates can be computed **concurrently**, breaking the sequential bottleneck. This iterative process continues until a suitable stopping criterion is met, such as the norm of the difference between consecutive iterates $\|\boldsymbol{z}_k^t - \boldsymbol{z}_k^{t-1}\|$ being sufficiently small. The whole process of applying Jacobi decoding in discrete autoregressive normalizing flow inference is summarized in Algorithm 1.

**Remark 3.1.** *Jacobi decoding techniques have been explored in other generative modeling contexts, such as language models and autoregressive image generation (Santilli et al., 2023; Song et al., 2021b). However, the straightforward application of Jacobi decoding has often shown limited success, e.g., marginal speedups*

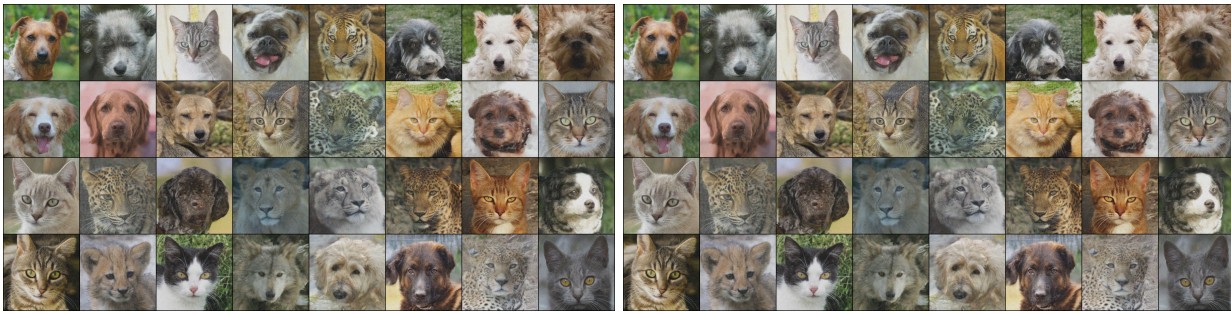

(a) Sequential            (b) Ours, 4.5 times acceleration

Figure 3: Visualization comparison on AFHQ. Our method accelerates generation by 4.5 times while maintaining the quality and fidelity of the generated content. Additional visual comparisons on CIFAR-10 and CIFAR-100 are available in Figure A7 and Figure A8.

*for language models (Kou et al., 2024) or compromised sample quality in image synthesis (Teng et al., 2025). Despite these challenging precedents, we hypothesize that discrete autoregressive normalizing flows possess characteristics that might mitigate such issues and make Jacobi iteration a more viable strategy. The empirically observed redundant dependencies suggest that the system might tolerate the use of slightly inaccurate information $z_{k,<l}^t$ from the previous iteration, which is fundamental to enabling Jacobi's parallel updates. Moreover, the deterministic nature of inversion avoids the compounding sampling errors present in stochastic models, and operating in continuous spaces could enable smoother iterative convergence than in discrete settings. These points provide a rationale for incorporating Jacobi decoding for discrete autoregressive normalizing flows.*

### 3.4 Convergence

In this section, we theoretically analyze the convergence of the proposed Jacobi iterative inference method and identify two crucial properties:

- The iteration exhibits local superlinear convergence under certain conditions, implying rapid convergence in practice.

- Due to the inherent triangular dependency structure of discrete autoregressive normalizing flow, the iteration is guaranteed to converge to the exact solution with no more than the number of steps of the original inference, providing a worst-case bound.

Firstly, we analyze the local convergence behavior. Assuming an appropriate initialization close to the true solution, the Jacobi iteration converges superlinearly.

**Proposition 3.1** (Superlinear Convergence Speed). *For the iterative sequence $z_k^t$ defined in Algorithm 1, $\exists \delta > 0$, s.t. $\forall z_k^0$ satisfies $\|z_k^0 - z_k\| < \delta$, the iterative sequence $\{z_k^t\}$ converges to $z_k$, with superlinear convergence rate, i.e., $\|z_k^{t+1} - z_k\| = o(\|z_k^t - z_k\|)$.*

Appendix B provides the detailed proof. Beyond the convergence rate, the specific structure of discrete autoregressive normalizing flow inference provides a strong global guarantee. The task is equivalent to solving a triangular system of equations, where the $l$-th unknown sub-variable depends only on preceding unknowns. This structural property, also noted in Santilli et al. (2023); Song et al. (2021b), ensures that the Jacobi method converges to the exact solution in at most $L$ iterations for a sequence with length $L$.

**Proposition 3.2** (Finite Convergence Guarantee). *For the iterative sequence $z_k^t$ defined in Algorithm 1. Denoting the sequence length of $z_k$ as L, we have $z_k^L = z_k$.*

Table 1: Comparison of sequential inference, uniform Jacobi decoding, and our approach. The subscript indicates the maximum deviation in three runs.

| Configuration | | Generation Speed | | Generation Quality | | |
|---|---|---|---|---|---|---|
| Dataset | Method | Time (s) ↓ | Speed Up ↑ | FID ↓ | CLIP-IQA ↑ | BRISQUE ↑ |
| CIFAR-10 | Sequential | $9.56_{\pm 0.42}$ | $1.0\times$ | $9.71_{\pm 0.03}$ | $0.35_{\pm 0.00}$ | $56.35_{\pm 0.21}$ |
| | UJD | $3.92_{\pm 0.09}$ | $2.4\times$ | $10.19_{\pm 0.03}$ | $0.35_{\pm 0.00}$ | $56.79_{\pm 0.20}$ |
| | **Ours** | $\mathbf{2.63}_{\pm 0.13}$ | $\mathbf{3.6\times}$ | $10.20_{\pm 0.03}$ | $0.35_{\pm 0.00}$ | $56.78_{\pm 0.19}$ |
| CIFAR-100 | Sequential | $9.57_{\pm 0.27}$ | $1.0\times$ | $8.22_{\pm 0.03}$ | $0.35_{\pm 0.00}$ | $57.75_{\pm 0.12}$ |
| | UJD | $3.30_{\pm 0.10}$ | $2.9\times$ | $8.26_{\pm 0.10}$ | $0.35_{\pm 0.00}$ | $57.76_{\pm 0.06}$ |
| | **Ours** | $\mathbf{2.04}_{\pm 0.03}$ | $\mathbf{4.7\times}$ | $8.19_{\pm 0.16}$ | $0.35_{\pm 0.00}$ | $57.78_{\pm 0.12}$ |
| AFHQ | Sequential | $186.28$ | $1.0\times$ | $15.42$ | $0.63$ | $15.55$ |
| | UJD | $219.24$ | $0.8\times$ | $15.44$ | $0.63$ | $15.44$ |
| | **Ours** | $\mathbf{41.21}$ | $\mathbf{4.5\times}$ | $15.44$ | $0.63$ | $15.56$ |

This finite convergence occurs because the triangular structure allows information to propagate definitively through the sequence during the updates. Appendix B presents a formal proof. Proposition 3.2 thus provides a strict upper bound on the computational steps needed to reach the exact solution.

These propositions jointly establish the inherent efficiency of Jacobi decoding. Proposition 3.1 reveals that the error reduces with a superlinear rate. This indicates the convergence can be fast, as each iteration $t < L$ yields an increasingly substantial reduction in their respective errors (Dennis & Schnabel, 1996). Note that though we assume a close initialization in the theoretical analysis, the convergence speed remains fast under various initializations, as shown in Section 4.3. On the other hand, Proposition 3.2 further presents the convergence to the exact solution $z_k$ is guaranteed in at most $L$ iterations. These results confirm the potential of applying Jacobi decoding for fast inference in discrete autoregressive normalizing flows.

### 3.5 Where to Use Jacobi Decoding

While the theoretical analysis shows promise for Jacobi iteration, its uniform application involves practical trade-offs. Concretely, in one iteration, we need to update all $L$ elements of the sequence simultaneously, which trades off memory for the time required per iteration in the original sequential setting, where only one sequence element is updated. Moreover, the commonly used key-value (KV) cache for optimized attention operators in modern transformer architectures (Ott et al., 2019) is not directly applicable to Jacobi decoding, as the decoded sub-variables are approximated and must be updated at each iteration. These limitations are particularly relevant where dependencies are strong, potentially making the uniform Jacobi decoding approach slower than optimized sequential decoding in such scenarios.

Motivated by this trade-off and the depthwise heterogeneity of sequential redundancy presented in Section 3.2, we exploit that the first ($K$-th) layer often exhibits stronger dependencies and propose our selective layer processing strategy. On modern transformer architectures with a KV cache, it uses standard sequential decoding for the dependency-heavy first layer, where the original sequential decoding works well. Subsequently, it switches to the parallel Jacobi iteration for the remaining layers, where higher redundancy is expected. It enables the benefits of parallelism while avoiding the high additional computational costs in the first layer due to stronger dependencies. This application aims to achieve more effective overall inference acceleration by strategically applying the Jacobi decoding method to layers with higher redundancy.

## 4 Experiments

In this section, we apply our approach to TarFlow, a state-of-the-art discrete autoregressive normalizing flow model (Zhai et al., 2025). Experiments are conducted on CIFAR-10 and CIFAR-100 (Krizhevsky, 2009) using models trained from scratch, and AFHQ (Choi et al., 2020) with a resolution of 256×256 using the released TarFlow checkpoint. The compared baselines include the standard sequential inference, the default inference method for discrete autoregressive normalizing flows, and the uniform Jacobi decoding (UJD) method, in

which the Jacobi decoding strategy is applied to all layers. The default stopping threshold $\tau$ for Jacobi iterations is set as 0.5. More experimental details, including network architectures and hyperparameters, are available in Appendix E.1.

Our main evaluation covers both computational efficiency and the quality of the generated samples. Generation speed is quantified by the average inference time per batch and the overall speedup relative to the sequential baseline, both measured on two L40S GPUs. For generative quality, we utilize the Fréchet Inception Distance (FID) (Heusel et al., 2017), a widely adopted metric that measures the perceptual similarity between the distribution of generated images and the real data distribution. For perceptual quality, we report two more no-reference metrics: CLIP-IQA (Wang et al., 2023), which assesses quality based on alignment with CLIP embeddings, and BRISQUE (Mittal et al., 2012), a blind image quality assessor sensitive to common distortions.

For comprehensiveness, we additionally test on smaller-scale Masked Autoregressive Flows (MAFs) (Papamakarios et al., 2017) on both image generation and Boltzmann distribution approximation tasks, as detailed in Appendix E.3. The results also confirm that our method achieves significant acceleration and further verify its generality.

## 4.1 Comparisons on Inference

We evaluate our method against the standard sequential baseline and the UJD method. As illustrated in Figure 3 and Appendix E, visual inspection of samples confirms that our method maintains high perceptual fidelity, producing outputs visually comparable to the original sequential generations. This qualitative observation is confirmed by quantitative analysis detailed in Table 1. Metrics such as FID, CLIP-IQA, and BRISQUE indicate that generative quality is largely preserved with both UJD and our method across all scenarios, with only small degradation relative to the sequential baseline.

However, while UJD demonstrates acceleration benefits on the smaller CIFAR-10 and CIFAR-100 datasets, it fails on the larger AFHQ dataset, resulting in inference speed slower than the sequential baseline. This poorer performance on AFHQ is likely attributable to higher per-iteration computational costs, combined with potentially stronger dependencies that negate parallel gains. Conversely, our method consistently achieves substantial speedups across all datasets by selectively applying Jacobi iterations. Notably, our method achieves up to a 4.7-times acceleration compared to the sequential baseline, as shown in Table 1, highlighting its effectiveness. This confirms that the proposed selective approach is crucial for achieving effective and generalizable acceleration without sacrificing generation quality.

## 4.2 Verification of Analysis

To experimentally validate our theoretical analysis and empirical observations, we analyze the convergence dynamics of the Jacobi iterations. Figure 4 plots the error measured as the $\ell_2$ norm of the difference between the iterate $\boldsymbol{z}_k^t$ and the ground truth $\boldsymbol{z}_k$ from sequential inference during the iteration process across different layers on the AFHQ dataset. As a reference, we also present the error variation of the original sequential inference, where the un-inferred sub-variables are regarded as the input sub-variables $\boldsymbol{z}^{t+1}$ according to the default implementation for calculation. The results clearly show a rapid decrease in error for the Jacobi process, often reaching nearly zero error in substantially fewer iterations than the

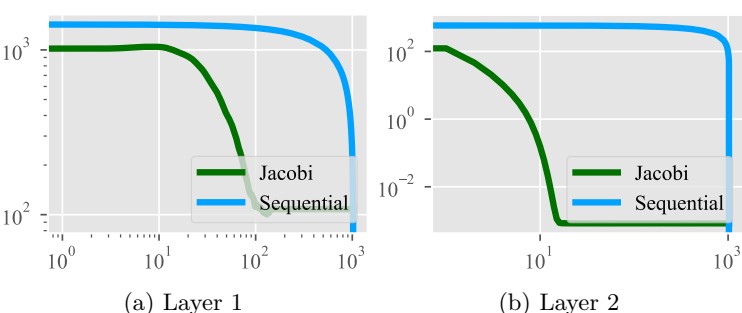

(a) Layer 1      (b) Layer 2

Figure 4: Convergence dynamics of Jacobi decoding across the first two network layers. Full results for all layers are shown in Figure A2. The plot shows the variation in the error (measured by the $\ell_2$ norm of the difference between the current iterate and the sequential output) over iterations, demonstrating fast overall convergence and the notably slower convergence of the first layer.

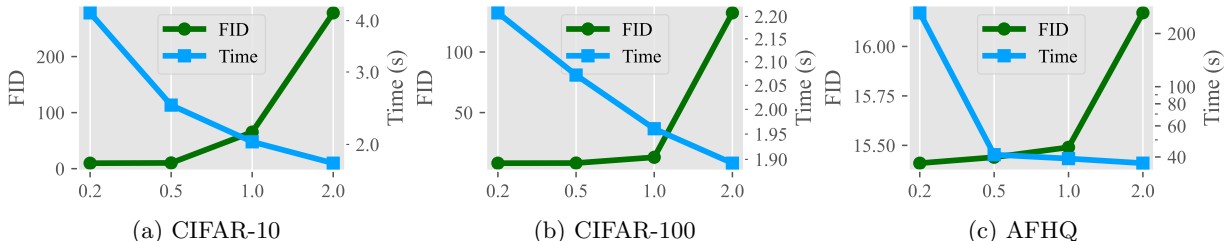

Figure 5: Ablation study on the stopping threshold $\tau$: FID scores and inference times for our method across different $\tau$ values, illustrating the speed-quality trade-off.

theoretical worst-case bound $L$, providing empirical support for its fast-convergence properties. Furthermore, Figure 4 reveals distinct convergence behavior across layers. The error associated with the first layer decreases noticeably more slowly via Jacobi iterations than that of subsequent layers. This directly validates our observation in Section 3.2 of stronger dependencies in the initial layer and empirically confirms the rationale behind the selective strategy, which applies parallel iterations to the faster-converging later layers. These results confirm the layer-wise differences in dependencies, thereby verifying the effectiveness of our method.

### 4.3 Ablation Study

**Influence of $\tau$.** To further understand the impact of the stopping threshold hyperparameter $\tau$, we perform an ablation study. We vary the value of $\tau$ and measure the resulting generative quality by FID and inference time. The results, illustrating the trade-off between these two metrics, are presented in Figure 5. As expected, increasing the threshold $\tau$ allows parallel iterations to terminate earlier, thereby significantly reducing the overall inference time. However, allowing larger differences between consecutive iterates before stopping can lead to a less precise generation. This is reflected in the FID scores, which tend to increase as $\tau$ becomes larger. Notably, the results show that for values $\tau$ below 1.0, the increase in FID is relatively gradual, while the reduction in inference time remains substantial. This supports that, with an appropriately chosen $\tau$, our method effectively increases generation speed with only a minor impact on generation quality. $\tau = 0.5$ consistently provides a favorable balance, achieving considerable acceleration while maintaining generative quality close to the baseline. Therefore, we adopt $\tau = 0.5$ as the default setting for all other experiments presented in this paper.

**Influence of initialization.** We perform an ablation study with various initialization methods of $z_k^0$, including zero initialization $z_k^0 = \mathbf{0}$, standard normal initialization $z_k^0 \sim \mathcal{N}(\mathbf{0}, I)$, and initialization with the output of previous layer $z_k^0 = z_{k+1}$. Experimental results in Figure 6 show that the acceleration performance remains similar across various initializations, supporting our analysis of superlinear convergence speed as general and insensitive to specific initialization methods.

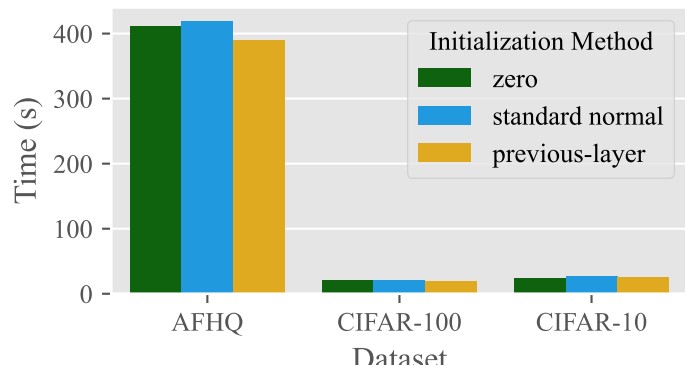

Figure 6: Ablation study on different initializations.

## 5 Conclusion

In this paper, we first observed the dependency redundancy within the autoregressive normalizing flow model and its significant variation across different layers. Based on this observation, we propose selectively applying the parallel Jacobi decoding method to layers with high dependency redundancy to accelerate inference. Theoretical analysis demonstrated the superlinear convergence of the proposed approach and provided a

worst-case guarantee on the total number of required iterations. Comprehensive experiments verified the correctness of the theoretical analysis and demonstrated that our method achieves significant inference acceleration across multiple scenarios, enhancing the practical value of normalizing flow models.

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

## A   Notation

To promote readability, we present a table of notations below.

Table A1: Table of notations.

| Notation | Description |
| --- | --- |
| $\boldsymbol{x}$ | Vector of observed data. |
| $\boldsymbol{z}_k$ | Vector of $k$-th step random variable of a discrete normalizing flow. |
| $\boldsymbol{z}_{k,l}$ | $l$-th sub-variable in the split of $k$-th step random variable. |
| $\boldsymbol{z}_{k,l}^t$ | $l$-th sub-variable in the $k$-th step random variable at $t$-th decoding step. |
| $\mathbf{J}_f$ | Jacobian matrix of function $f$ |
| $T$ | Number of decoding steps in Jacobi decoding. |
| $L$ | Number of sub-variables in the split random variable. |
| $K$ | Number of steps in a discrete normalizing flow. |

## B   Theoretical Proofs

We first formally redefine our Jacobi iteration map as the function $F(\cdot)$.

**Definition B.1** (Jacobi Iteration Map). *Let $\boldsymbol{z}_{k+1}$ be a given vector sequence of length $L$, and let $s_k, g_k$ be given functions. The iteration map $F$ is defined component-wise for $\boldsymbol{z}$ as:*

$$F(\boldsymbol{z})_l = \begin{cases} \boldsymbol{z}_{k+1,1} & l = 1 \\ \boldsymbol{z}_{k+1,l} \odot \exp\big(-s_k(\boldsymbol{z}_{<l})\big) + g_k(\boldsymbol{z}_{<l}) & l = 2, \ldots, L \end{cases} \tag{9}$$

*where $\boldsymbol{z}_{<l} := [z_1, \ldots, z_{l-1}]^T$. The iterative sequence is generated by $\boldsymbol{z}_k^{t+1} = F(\boldsymbol{z}_k^t)$ with initial $\boldsymbol{z}_k^0$.*

It is easy to observe that there exists a fixed point $\boldsymbol{z}_k^*$ for this iteration. For example, the output $\boldsymbol{z}_k$ from the sequential decoding approach in equation 5 is a fixed point. Moreover, as we use neural networks as parameterized functions $s_k$ and $g_k$, the map $F$ is continuously differentiable. Under these observations, we have the iterative sequence $\boldsymbol{z}_k^t$ converges to $\boldsymbol{z}_k^*$ with at least a superlinear convergence rate starting from a close initial sequence, as shown in the following theorem.

**Proposition B.1** (Superlinear Convergence Rate). *There exists $\delta > 0$ such that if $\left\| \boldsymbol{z}_k^0 - \boldsymbol{z}_k^* \right\| < \delta$, the iterative sequence $\{\boldsymbol{z}_k^t\}$ converges to $\boldsymbol{z}_k^*$ with at least a superlinear convergence rate. This means that the error satisfies:*

$$\|\boldsymbol{z}_k^{t+1} - \boldsymbol{z}_k^*\| = o(\|\boldsymbol{z}_k^t - \boldsymbol{z}_k^*\|) \quad as\ \boldsymbol{z}_k^t \to \boldsymbol{z}_k^*. \tag{10}$$

*Proof.* Denote $\boldsymbol{e}^t = \boldsymbol{z}_k^t - \boldsymbol{z}_k^*$ the approximation error at iteration $t$.

The structure of the iteration map $F(\boldsymbol{z})$ (as detailed in prior definitions: $F(\boldsymbol{z})_1 = \boldsymbol{z}_{k+1,1}$ is constant, and $F(\boldsymbol{z})_l = F(\boldsymbol{z}_{<l})$ for $l = 2, \ldots, L$) ensures that the Jacobian matrix $\mathbf{J}_F(\boldsymbol{z})$ is strictly lower triangular for any $\boldsymbol{z}$. This implies that $\mathbf{J}_F(\boldsymbol{z}_k^*)$ is strictly lower triangular. Consequently, all eigenvalues of $\mathbf{J}_F(\boldsymbol{z}_k^*)$ are zero. Therefore, the spectral radius of the Jacobian at the fixed point is

$$\rho(\mathbf{J}_F(\boldsymbol{z}_k^*)) = 0. \tag{11}$$

As $F$ is continuously differentiable, Taylor's theorem allows us to expand $F(\boldsymbol{z}_k^t)$ around $\boldsymbol{z}_k^*$. For $\boldsymbol{z}_k^t$ sufficiently close to $\boldsymbol{z}_k^*$:

$$F(\boldsymbol{z}_k^t) = F(\boldsymbol{z}_k^*) + \mathbf{J}_F(\boldsymbol{z}_k^*)(\boldsymbol{z}_k^t - \boldsymbol{z}_k^*) + o(\left\| \boldsymbol{z}_k^t - \boldsymbol{z}_k^* \right\|). \tag{12}$$

Using the iteration definition $\boldsymbol{z}_k^{t+1} = F(\boldsymbol{z}_k^t)$ and the property at fixed-point $\boldsymbol{z}_k^* = F(\boldsymbol{z}_k^*)$, we have:

$$\boldsymbol{z}_k^{t+1} - \boldsymbol{z}_k^* = F(\boldsymbol{z}_k^t) - F(\boldsymbol{z}_k^*) = \mathbf{J}_F(\boldsymbol{z}_k^*)(\boldsymbol{z}_k^t - \boldsymbol{z}_k^*) + o(\left\| \boldsymbol{z}_k^t - \boldsymbol{z}_k^* \right\|). \tag{13}$$

This yields an error propagation dynamic:

$$\boldsymbol{e}^{t+1} = \mathbf{J}_F(\boldsymbol{z}_k^*)\boldsymbol{e}^t + o(\|\boldsymbol{e}^t\|). \tag{14}$$

Standard theorems on iterative methods (e.g., results related to Q-order of convergence as found in previous work (Ortega & Rheinboldt, 2000), see discussion around Theorem 10.1.4 for Q-superlinear) state that if an iteration converges. Its error satisfies the relationship in equation 14, then the convergence is Q-superlinear if and only if $\rho(\mathbf{J}_F(\boldsymbol{z}_k^*)) = 0$. Meanwhile, the condition $\rho(\mathbf{J}_F(\boldsymbol{z}_k^*)) < 1$ (which is satisfied here since $\rho = 0$) ensures that $\boldsymbol{z}_k^*$ is a point of attraction, so for $\boldsymbol{z}_k^0$ sufficiently close to $\boldsymbol{z}_k^*$, the sequence $\{\boldsymbol{z}_k^t\}$ is guaranteed to converge at $\boldsymbol{z}_k^*$.

Given that $\rho(\mathbf{J}_F(\boldsymbol{z}_k^*)) = 0$, the cited convergence theory directly implies that the iteration is Q-superlinear. By definition, Q-superlinear convergence means that $\|\boldsymbol{e}^{t+1}\| = o(\|\boldsymbol{e}^t\|)$ as $\boldsymbol{e}^t \to \boldsymbol{0}$. Therefore, we conclude that:

$$\|\boldsymbol{z}_k^{t+1} - \boldsymbol{z}_k^*\| = o(\|\boldsymbol{z}_k^t - \boldsymbol{z}_k^*\|) \quad \text{as } \boldsymbol{z}_k^t \to \boldsymbol{z}_k^*. \tag{15}$$

This demonstrates at least a superlinear convergence rate. □

**Proposition B.2** (Finite Convergence Guarantee). *For iteration map $F$ and iterative sequence $\{\boldsymbol{z}_k^t\}_{t\geq 0}$ defined in Definition B.1, the iteration converges to the fixed point in at most $L$ steps:*

$$\boldsymbol{z}_k^t = \boldsymbol{z}_k^* \quad \forall t \geq L. \tag{16}$$

*Proof.* The core property is that the $l$-th component of the output, $(F(\boldsymbol{z}))_l$, depends only on the first $l-1$ components of the input $\boldsymbol{z}$, specifically $\boldsymbol{z}_{<l}$. We also know that $(F(\boldsymbol{z}_k^0))_1 = \boldsymbol{z}_{k+1,1} = \boldsymbol{z}_{k,1}^*$.

We will prove by induction on the iteration step $t$ (from $t = 1$ to $t = L$) that the first $t$ components of the iterate $\boldsymbol{z}_k^t$ match those of the fixed point $\boldsymbol{z}_k^*$. Let $P(t)$ be the statement:

$$P(t): \quad \boldsymbol{z}_{k,l}^t = \boldsymbol{z}_{k,l}^* \quad \forall 1 \leq l \leq t. \tag{17}$$

It is easy to check that the statement $P(1)$ holds. By assuming that $P(t)$ holds, that is, $\boldsymbol{z}_{k,l}^t = \boldsymbol{z}_{k,l}^*$ for all $1 \leq l \leq t$, we want to show that $P(t+1)$ holds, meaning $\boldsymbol{z}_{k,l}^{t+1} = \boldsymbol{z}_{k,l}^*$ for all $1 \leq l \leq t+1$.

Since $\boldsymbol{z}_k^{t+1} = F(\boldsymbol{z}_k^t)$:

- **For $1 \leq l \leq t$:** The calculation of $\boldsymbol{z}_{k,l}^{t+1} = (F(\boldsymbol{z}_k^t))_l$ depends only on $\boldsymbol{z}_{k,<l}^t$. Given $j < l \leq t$, the inductive hypothesis $P(t)$ implies $\boldsymbol{z}_{k,j}^t = \boldsymbol{z}_{k,j}^*$ for these components. Thus, $\boldsymbol{z}_{k,<l}^t = \boldsymbol{z}_{k,<l}^*$. Because $(F(\cdot))_l$ only depends on these first $l-1$ components, we have

$$(F(\boldsymbol{z}_k^t))_l = (F(\boldsymbol{z}_k^*))_l. \tag{18}$$

  Since $\boldsymbol{z}_k^*$ is a fixed point hence $F(\boldsymbol{z}_k^*) = \boldsymbol{z}_k^*$, we have

$$\boldsymbol{z}_{k,l}^{t+1} = (F(\boldsymbol{z}_k^*))_l = \boldsymbol{z}_{k,l}^*. \tag{19}$$

- **For $l = t+1$:** The calculation of $\boldsymbol{z}_{k,t+1}^{t+1} = (F(\boldsymbol{z}_k^t))_{t+1}$ depends only on $\boldsymbol{z}_{k,<t+1}^t$. The components in this sub-vector are $\boldsymbol{z}_{k,j}^t$ for $j = 1, \ldots, t$. By the inductive hypothesis $P(t)$, these are equal to the corresponding components of $\boldsymbol{z}_k^*$. Therefore, $\boldsymbol{z}_{k,<t+1}^t = \boldsymbol{z}_{k,<t+1}^*$. Because $(F(\cdot))_{t+1}$ only depends on the first $t$ components

$$(F(\boldsymbol{z}_k^t))_{t+1} = (F(\boldsymbol{z}_k^*))_{t+1} \tag{20}$$

  Using the fixed-point property:

$$\boldsymbol{z}_{k,t+1}^{t+1} = (F(\boldsymbol{z}_k^*))_{t+1} = \boldsymbol{z}_{k,t+1}^*. \tag{21}$$

  This indicates that the $(t+1)$-th component becomes correct at step $t+1$.

Combining equation 19 and equation 21, we show that $z_{k,l}^{t+1} = z_{k,l}^*, \forall 1 \le l \le t+1$. Thus, $P(t+1)$ holds.

By mathematical induction, $P(t)$ holds for all $t = 1, \ldots, L$. In particular, $P(L)$ holds:

$$z_{k,l}^L = z_{k,l}^* \quad \forall 1 \le l \le L. \tag{22}$$

This implies the entire vector is guaranteed to match the fixed point after $L$ steps:

$$z_k^L = z_k^*. \tag{23}$$

Assume $z_k^t = z_k^*$ for some $t \ge L$. Then in the next iteration:

$$z_k^{t+1} = F(z_k^t) = F(z_k^*) = z_k^* \tag{24}$$

For the same reason, if the sequence reaches $z_k^*$ at step $L$, it remains at $z_k^*$ for all subsequent steps. Therefore, it is shown that

$$z_k^t = z_k^* \quad \forall t \ge L. \tag{25}$$

$\square$

## C   Limitations and Future Work

While our method demonstrates promising improvements in inference acceleration for autoregressive normalizing flow models, it also highlights several open problems. First, although sequential and depthwise redundancy is commonly observed in trained models, it remains unknown whether this redundancy persists to the same extent in partially trained or undertrained models. This uncertainty might affect model effectiveness when models underfit or are in the early stages of training. Second, this paper focuses exclusively on inference-time acceleration. The potential for leveraging our principles to optimize the training process or guide neural architecture design has not been explored. Future work could further investigate the nature of these observed sequential and depthwise redundancies. Such insights might then be leveraged to guide model training strategies and inform architectural design, potentially leading to models that are inherently more efficient.

## D   Analysis on Memory Complexity

Both the original sequential inference and our method have $O(L^2)$ memory complexity. The original TarFlow requires $O(L^2)$ memory due to the attention mechanism operating on $L$-length sequences, with KV cache not adding to the asymptotic complexity. Similarly, our method maintains the same $O(L^2)$ complexity since we perform attention operations on the entire $L$-length sequence at each step, without requiring shared attention matrices across different steps.

While the asymptotic complexity is identical, the actual memory usage differs in practice. On AFHQ (batch size 16), our method uses only 5.2GB of memory compared to 7.8GB for the baseline implementation with KV cache. This difference arises because the baseline stores additional $K$ and $V$ tensors to avoid redundant computations, whereas our approach achieves acceleration through parallel processing without this storage overhead. Thus, our method demonstrates better memory efficiency despite processing $L$ inputs in parallel.

## E   Additional Experiments

### E.1   Experimental Details on TarFlow

**Model Details.** The network architectures for our main baseline models are adopted from the publicly available implementation of TarFlow[1]. For experiments on the AFHQ dataset, we utilize the pre-trained checkpoint released by the TarFlow authors. Due to computational resource constraints, training models on the ImageNet dataset according to the original TarFlow configurations was not feasible within the scope

---

[1]`https://github.com/apple/ml-tarflow`

of this work. For experiments on the CIFAR datasets, we largely follow the default settings provided by TarFlow, with a few adjustments. These modifications are implemented to better suit our experimental objectives or to accommodate resource limitations. Table A2 summarizes the key configurations for each dataset.

Table A2: Experimental configuration for each dataset.

|  | CIFAR-10 | CIFAR-100 | AFHQ |
|---|---|---|---|
| Resolution | $32 \times 32$ | $32 \times 32$ | $256 \times 256$ |
| Patch size $P$ | 2 | 2 | 8 |
| Sequence length $L$ | 256 | 256 | 1024 |
| Number of blocks $K$ | 6 | 6 | 8 |
| Layers per block | 6 | 6 | 8 |
| Hidden dimension | 256 | 256 | 768 |
| Batch size | 256 | 256 | 256 |

**Evaluation Details.** To estimate generation speed, we compute the average time per batch across 10 distinct runs. For Fréchet Inception Distance (FID) estimation, we compute the distance between the original dataset and a generated dataset of the same size, following the standard FID definition. To evaluate generation quality, we use metrics such as CLIP-IQA and BRISQUE, computing the average score for each metric over the set of generated samples, matching the original dataset size. These approaches, which involve averaging and large sample sizes, ensure stable, representative results. The evaluation methods employed, which involve averaging results across multiple batches and utilizing extensive datasets, can ensure representative results. It is consistent with established practices in the literature (Song et al., 2021b; Teng et al., 2025). For CIFAR datasets, we additionally report the maximum deviation in three runs. The magnitude of this deviation is considerably smaller than the performance differentials observed between methods, thereby affirming the statistical significance of our comparative results and the validity of the reported enhancements.

**Inference Details.** Table A3 reports the average number of iterations per layer during inference with our Selective Jacobi Decoding. Layer 1 uses standard sequential decoding ($L - 1$ steps), while the remaining layers use Jacobi iteration. Notably, almost all Jacobi layers converge in very few iterations (typically 4–7), far below the worst-case bound of $L$, validating our theoretical analysis. The relatively higher iteration count at Layer 2 on CIFAR-10 is consistent with our observation of depthwise heterogeneity, in which layers closer to the first layer tend to exhibit stronger sequential dependencies.

Table A3: Average number of Jacobi iterations per layer ($\tau = 0.5$). Layer 1 uses sequential decoding; remaining layers use Jacobi iteration.

| Layer | CIFAR-10 | CIFAR-100 | AFHQ |
|---|---|---|---|
| 1 (Sequential) | 255 | 255 | 1023 |
| 2 (Jacobi) | 53.9 | 7.5 | 6.6 |
| 3 (Jacobi) | 4.9 | 4.7 | 6.0 |
| 4 (Jacobi) | 4.0 | 4.0 | 5.2 |
| 5 (Jacobi) | 3.0 | 4.0 | 5.0 |
| 6 (Jacobi) | 6.1 | 5.2 | 5.9 |
| 7 (Jacobi) | — | — | 4.5 |
| 8 (Jacobi) | — | — | 4.0 |

Table A4 presents the per-layer runtime breakdown for both sequential inference and our method. In sequential inference, each layer takes approximately the same time. Under SJD, Layer 1 (sequential) dominates the total cost, while each Jacobi layer completes in a fraction of the time. This confirms that the acceleration stems from replacing expensive sequential decoding with fast-converging Jacobi iterations on layers with high redundancy.

Table A4: Per-layer runtime breakdown comparing sequential inference and SJD. "Other" includes self-denoising, inter-GPU communication, and noise generation overhead, etc.

| | Sequential | | SJD (Ours) | | |
| Layer | Time (s) | % | Time (s) | % | Jacobi Iters |
|---|---|---|---|---|---|
| *CIFAR-10* | | | | | |
| 1 (Seq) | 1.45 | 14.8% | 1.50 | 55.7% | 255 |
| 2 (Jacobi) | 1.45 | 14.8% | 0.69 | 25.7% | 53.9 |
| 3–6 (Jacobi) | 5.82 | 59.4% | 0.23 | 8.6% | 3.0–6.1 |
| Other | 1.08 | 11.0% | 0.27 | 10.0% | — |
| **Total** | **9.79** | **100%** | **2.69** | **100%** | **3.6×** |
| *CIFAR-100* | | | | | |
| 1 (Seq) | 1.51 | 16.5% | 1.55 | 76.6% | 255 |
| 2 (Jacobi) | 1.50 | 16.4% | 0.10 | 4.7% | 7.5 |
| 3–6 (Jacobi) | 6.00 | 65.6% | 0.23 | 11.4% | 4.0–5.2 |
| Other | 0.15 | 1.6% | 0.15 | 7.2% | — |
| **Total** | **9.15** | **100%** | **2.02** | **100%** | **4.5×** |
| *AFHQ* | | | | | |
| 1 (Seq) | 21.60 | 12.5% | 21.46 | 52.0% | 1023 |
| 2 (Jacobi) | 20.85 | 12.1% | 2.56 | 6.2% | 6.6 |
| 3–8 (Jacobi) | 125.10 | 72.4% | 11.85 | 28.7% | 4.0–6.0 |
| Other | 5.35 | 3.1% | 5.38 | 13.1% | — |
| **Total** | **172.90** | **100%** | **41.25** | **100%** | **4.2×** |

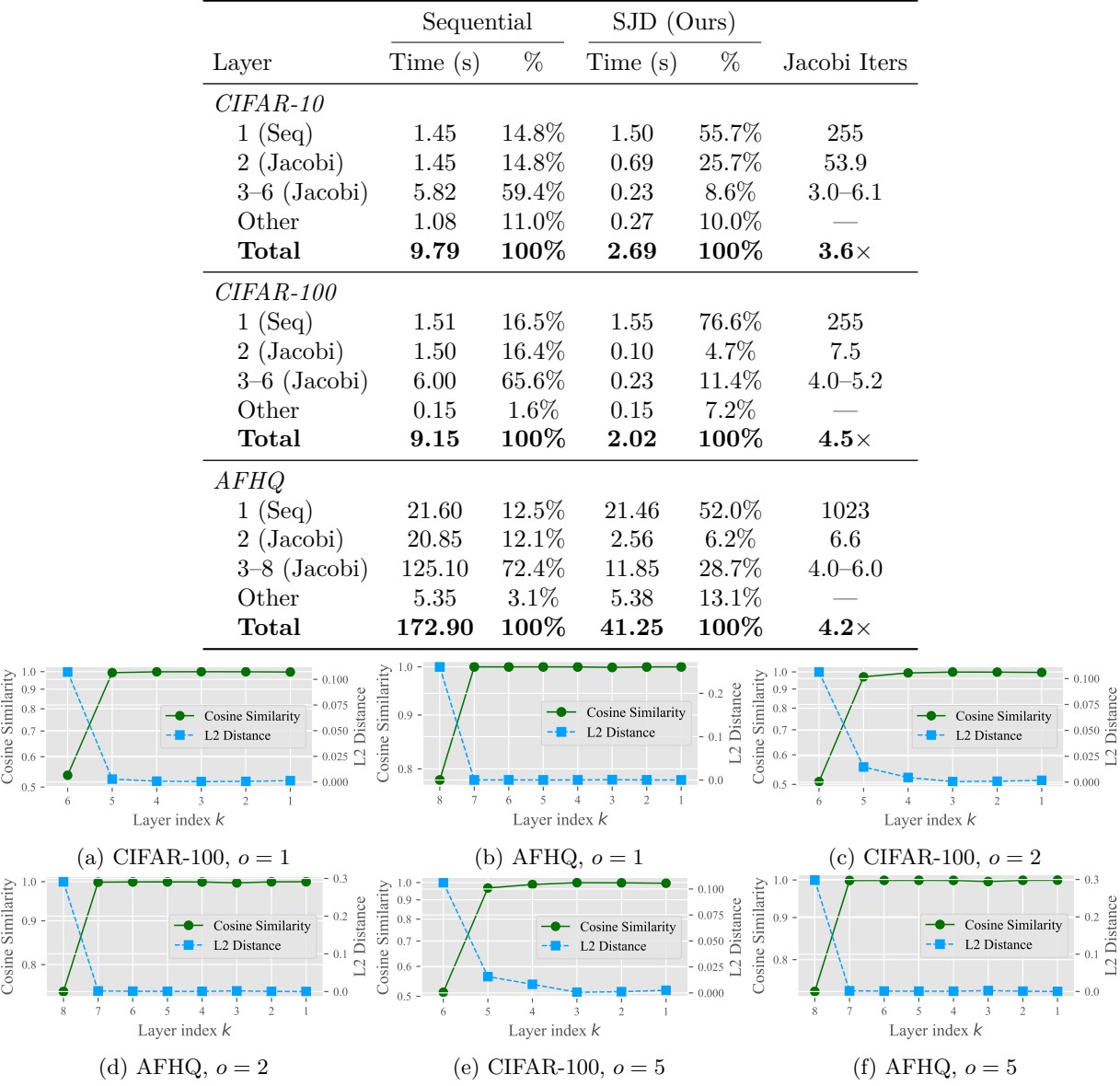

(a) CIFAR-100, $o = 1$  (b) AFHQ, $o = 1$  (c) CIFAR-100, $o = 2$

(d) AFHQ, $o = 2$  (e) CIFAR-100, $o = 5$  (f) AFHQ, $o = 5$

Figure A1: Cosine similarities and L2 distances between layer outputs from standard inference and inference with $o = 1$, $o = 2$, and $o = 5$ nearest preceding dependencies masked.

## E.2 Full Results of Fig. 1 and Fig. 4.

We provide full results of Fig. 1 and Fig. 4 in Fig. A1 and Fig. A2, respectively.

## E.3 Experiments on Masked Autoregressive Flow

In this section, we test Masked Autoregressive Flow (MAF) models on both image generation and Boltzmann distribution approximation tasks. The implementation from `nflows`[2] is used. Note that KV-cache does not apply to this MLP-based architecture. Therefore, Jacobi decoding could accelerate across all layers, so

---

[2]`https://github.com/bayesiains/nflows/blob/master/nflows/transforms/autoregressive.py`

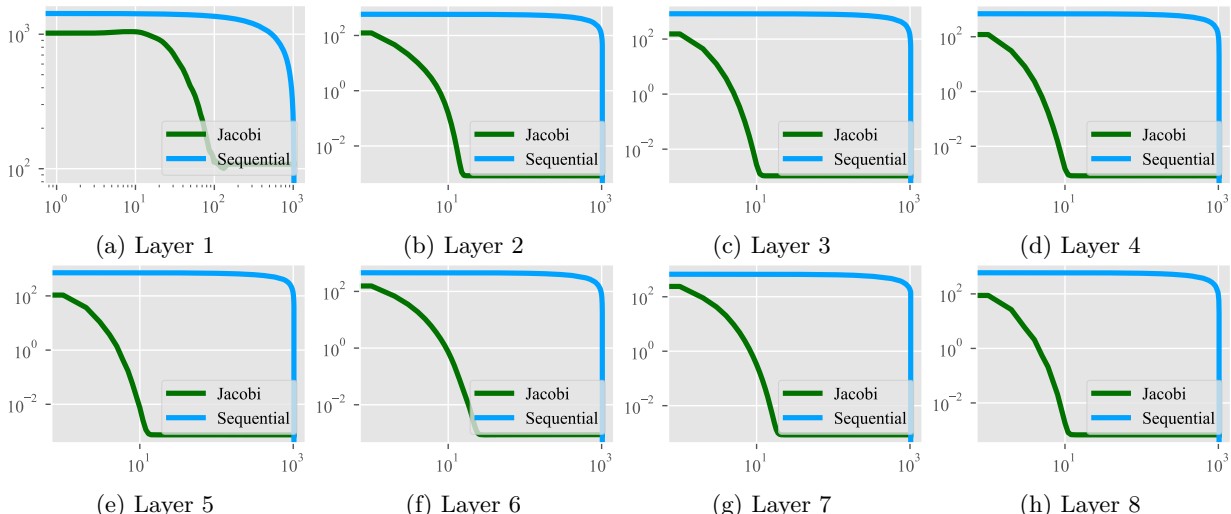

Figure A2: Convergence dynamics of Jacobi decoding across network layers. The plot shows the variation in the error (measured by the $\ell_2$ norm of the difference between the current iterate and the sequential output) over iterations, demonstrating fast overall convergence and the notably slower convergence of the first layer.

Table A5: Comparison between Sequential inference and our method of MAF on Boltzmann distribution approximation task.

| Method | Inference Time (s) | Average Energy / Site | Average Absolute Magnetization |
|---|---|---|---|
| Sequential | 16.84 | 0.0005 | 0.0500 |
| Ours | 1.07 | -0.0003 | 0.0498 |

we select all layers for Jacobi decoding rather than using it only on non-first layers, yielding even higher acceleration.

**Boltzmann Distribution Approximation.** We test on an 8-layer MAF trained via reverse KL to approximate the Boltzmann distribution for a high-temperature (T=3.0) disordered state of a 2D Ising model. The reverse KL divergence loss is reduced from an initial value of -1111 to a final value of -1129 over 3000 epochs. To evaluate the performance, we generated 100,000 samples and compared our method against the standard sequential method. The results demonstrate a significant acceleration with negligible impact on quality, as shown in Table A5.

The near-zero energy and magnetization values are consistent with disordered-state physics, confirming that sample quality is maintained. Notably, we achieve a 15.7x speedup. This experiment provides strong evidence of the general applicability of our method.

**Image Generation.** We conduct further experiments on an 8-layer Masked Autoregressive Flow (MAF) trained on binary MNIST. Due to the model's limited expressive power, the generative quality is low. However, we still observe that the generations from our method and sequential inference have very similar image quality, as shown in Fig. A3. To generate 100 images, our method takes only 15.24 seconds, while the original sequential method required 281.00 seconds. This is a significant 18.4x acceleration, providing strong evidence of our method's general applicability.

### E.4 Evaluation of Reconstruction Consistency

A core advantage of discrete autoregressive normalizing flows is their strictly invertible architecture, which allows for perfect reconstruction. To explicitly quantify any numerical deviation introduced by our parallel iterative approximation and to verify the preservation of the model's invertibility, we evaluate the reconstruction

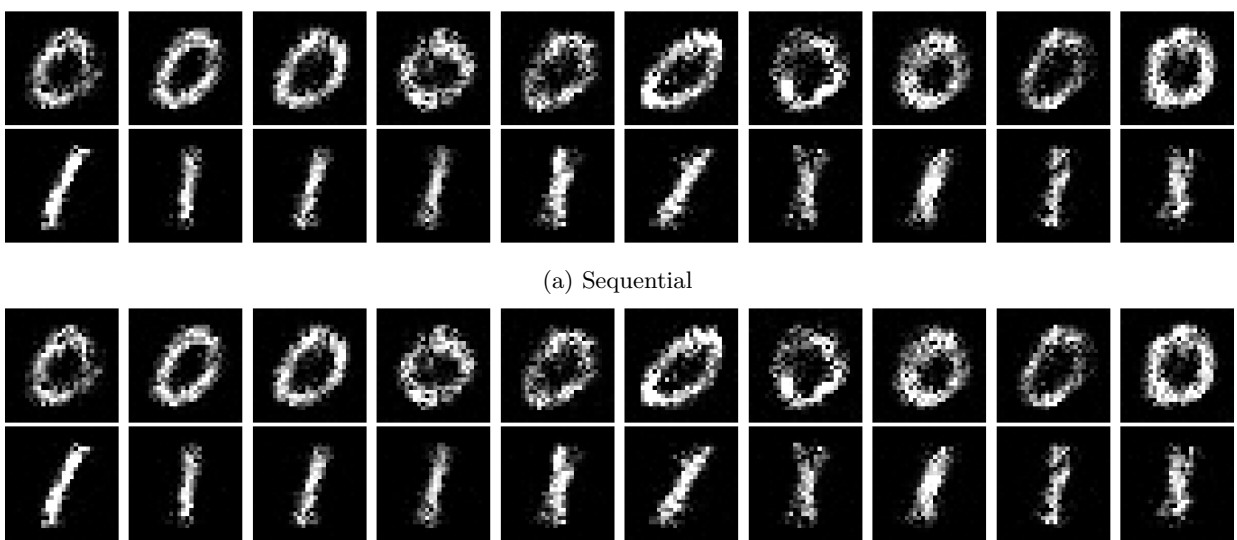

(a) Sequential

(b) Ours, 18.4 times acceleration

Figure A3: Visualization comparison on binary MNIST.

consistency. Specifically, we map real images $\boldsymbol{x}$ from the original dataset to the exact latent variables $\boldsymbol{z}$ using the standard sequential forward pass, and subsequently reconstruct them back to the pixel space $\hat{\boldsymbol{x}}$ using our SJD.

Our method achieves exceptionally low Mean Squared Error (MSE) scores between the original inputs $\boldsymbol{x}$ and the reconstructed outputs $\hat{\boldsymbol{x}}$: 0.00636 on CIFAR-10, 0.00313 on CIFAR-100, and 0.00122 on AFHQ (all evaluated with SJD, $\tau = 0.5$). These near-zero numerical errors confirm that the deviation introduced by relaxing the strict sequential dependency is virtually negligible, and that the parallel iterations converge tightly to the exact sequential solutions. This quantitative precision is further corroborated by visual inspection. As illustrated in Fig. A4, Fig. A5, and Fig. A6, the reconstructed images are visually indistinguishable from the original ones, with no perceptible loss of detail. Both the quantitative and qualitative results strongly verify that our method successfully preserves the strict bijective consistency and high-fidelity generation quality inherent to flow-based models.

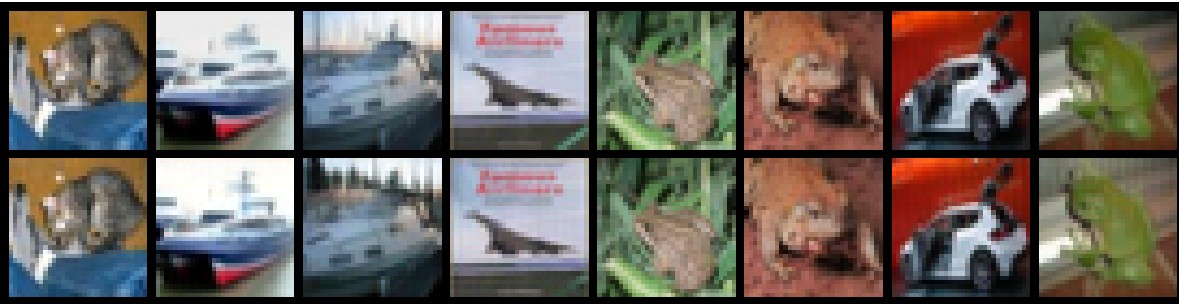

Figure A4: Reconstruction consistency on the CIFAR-10 test set. **Top row:** Original real images. **Bottom row:** Reconstructed images using our Selective Jacobi Decoding. The reconstructions are visually indistinguishable from the original inputs.

### E.5 Comparative Analysis with GAN and Diffusion Models

To contextualize the practical utility of our proposed acceleration method, we provide a comparative analysis against representative Generative Adversarial Networks (GANs) and Diffusion Models on the CIFAR-10

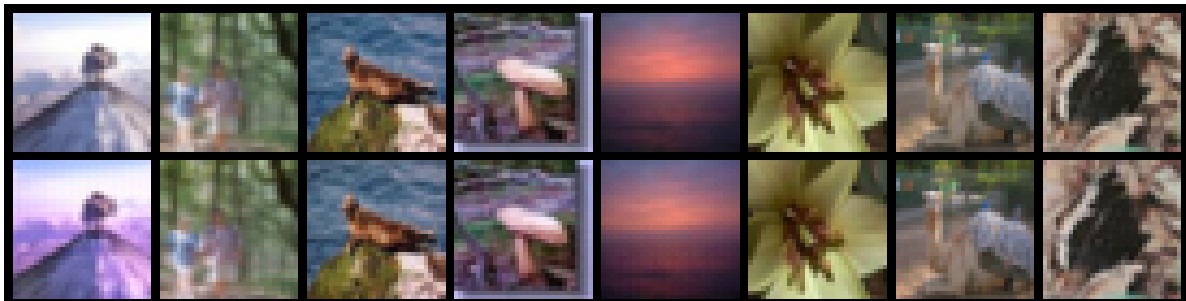

Figure A5: Reconstruction consistency on the CIFAR-100 test set. **Top row:** Original real images. **Bottom row:** Reconstructed images using our Selective Jacobi Decoding.

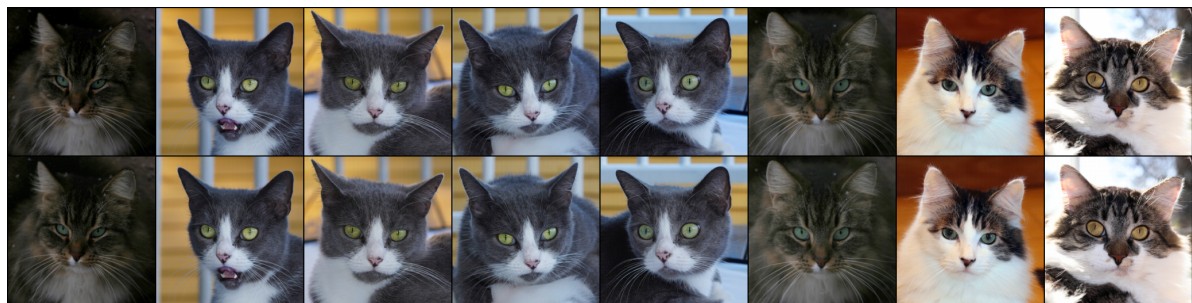

Figure A6: Reconstruction consistency on the AFHQ test set. **Top row:** Original real images. **Bottom row:** Reconstructed images using our Selective Jacobi Decoding.

dataset. For the GAN baseline, we trained FastGAN (Zhong et al., 2020) from scratch using the official implementation. For the diffusion baseline, we evaluated DDIM (Song et al., 2021a) using the publicly available `google/ddpm-cifar10-32` checkpoint at 20 inference steps to establish a comparable speed profile.

Table A6: Comparison of our method against FastGAN and DDIM on the CIFAR-10 dataset.

| Method | Inference Time (s) ↓ | FID ↓ |
|---|---|---|
| Fast GAN | 3.41 | 9.67 |
| DDIM (20 steps) | 2.31 | 19.32 |
| Ours | 2.63 | 10.20 |

As detailed in Table A6, our method demonstrates a highly competitive balance between generation speed and quality. It achieves faster inference than FastGAN with only a marginal trade-off in FID. While DDIM at 20 steps is slightly faster, it incurs a severe penalty to generation quality. Compared with DDIM at 20 steps, our method is only marginally slower but achieves substantially better generation quality.

### E.6 More visualized results

We provide more visualized experimental results on CIFAR-10 and CIFAR-100 in Fig. A7 and Fig. A8. All results consistently confirm the little impact of our method on generation quality.

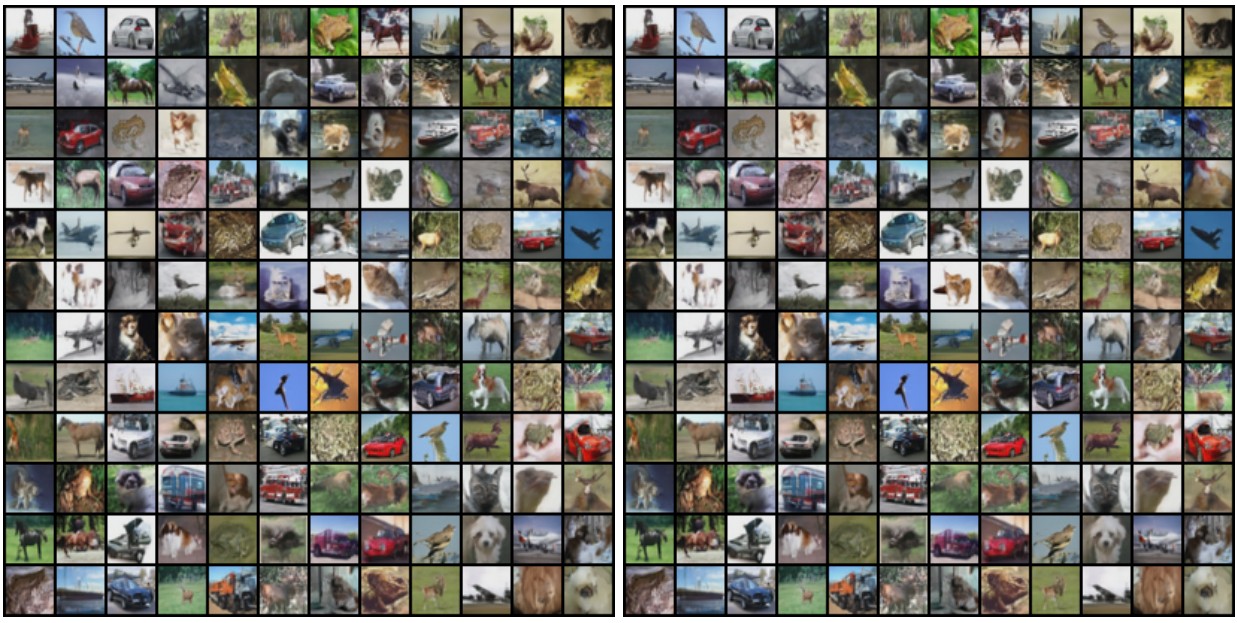

(a) Sequential

(b) Ours, 3.6 times acceleration

Figure A7: Visualization comparison on CIFAR-10.

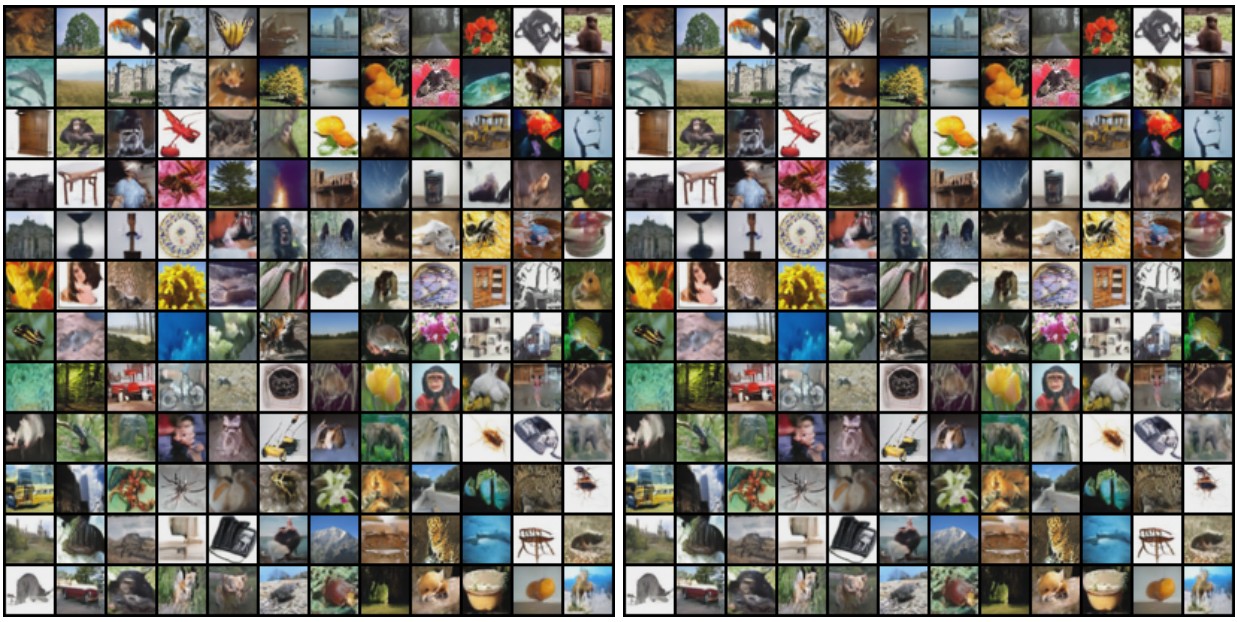

(a) Sequential

(b) Ours, 4.7 times acceleration

Figure A8: Visualization comparison on CIFAR-100.

