# OpenReview forum: "Accelerating Inference of Discrete Autoregressive Normalizing Flows by Selective Jacobi Decoding"
_TMLR — Accepted by TMLR_

### Review · Reviewer_tnWx · 2026-02-27

**Summary Of Contributions:**

This paper studies inference-time acceleration for discrete autoregressive normalizing flows (instantiated mainly on TarFlow). The key observation is that strict sequential dependence in the inverse/autoregressive sampling procedure is partly redundant, and that this redundancy is layer-dependent (stronger dependence in the first layer, more redundancy in later layers). Based on this, the authors propose Selective Jacobi Decoding (SJD): keep sequential decoding in the dependency-heavy first layer, and apply parallel Jacobi iterations in later layers to break sequential bottlenecks. They provide theoretical claims of superlinear local convergence and a finite-step bound (≤ sequence length L) due to the triangular structure, and report speedups up to 4.7× while maintaining similar quality on CIFAR-10/100 and AFHQ (256×256).

**Audience:**

Yes

**Audience Explanation:**

TMLR readers interested in generative modeling, normalizing flows, and efficient inference / decoding would likely care about this. The paper addresses a real practical drawback of autoregressive flows (sequential inversion bottleneck) and adapts a general acceleration idea (Jacobi/parallel fixed-point style decoding) in a way that is training-free and layer-aware, with meaningful speedups.

**Claims And Evidence:**

Yes

**Claims Explanation:**

1. The paper provides empirical redundancy evidence by testing masking nearest o dependencies during inference and showing layerwise deviation patterns via cosine similarity/L2 distance.

2. The paper provides acceleration results vs. baselines: Table 1 compares sequential, uniform Jacobi decoding (UJD), and the selective method. The selective method is substantially faster (e.g., up to 4.7× on CIFAR-100; 4.5× on AFHQ), while FID/CLIP-IQA/BRISQUE change only marginally. The UJD failure mode on AFHQ (slower than baseline) helps motivate “selective” rather than “always Jacobi.”

3. The authors also provide theoretical convergence discussion.

**Requested Changes:**

1. While the reported speedups are promising, the experimental setup would benefit from clearer reporting of key implementation details. In particular, it would be helpful to explicitly state the sequence length, batch size, and the average number of Jacobi iterations used per layer during inference. Additionally, a brief runtime breakdown (e.g., per-layer or per-stage cost) would help readers better understand where the acceleration is coming from and how it scales in practice.

2. Since one of the core advantages of flow-based models is their tractable likelihood, the evaluation would be strengthened by including likelihood-related metrics (e.g., bits-per-dimension or reconstruction consistency) under the accelerated decoding procedure, if applicable.

---

> ### Author Response · Authors · 2026-04-02
> **Response to Reviewer tnWx (1/2)**
>
> We sincerely appreciate the time and effort you have spent providing insightful feedback on our paper. We have carefully considered each of your detailed comments and have addressed them one by one to alleviate your concerns.
>
> > While the reported speedups are promising, the experimental setup would benefit from clearer reporting of key implementation details. In particular, it would be helpful to explicitly state the sequence length, batch size, and the average number of Jacobi iterations used per layer during inference. Additionally, a brief runtime breakdown (e.g., per-layer or per-stage cost) would help readers better understand where the acceleration is coming from and how it scales in practice.
>
> **Implementation Details.** Thank you for this constructive suggestion. As mentioned in Appendix E.1, we have discussed the detailed settings like the sequence length and batch size. To make this clearer, the following Table R1 summarizes the key configuration for each dataset. We have updated Appendix E.1 with this table to make the implementation details more explicit.
>
> *Table R1: Experimental configuration for each dataset.*
> | | CIFAR-10 | CIFAR-100 | AFHQ |
> |---|---|---|---|
> | Resolution | 32×32 | 32×32 | 256×256 |
> | Patch size $P$ | 2 | 2 | 8 |
> | Sequence length $L$ | 256 | 256 | 1024 |
> | Number of blocks $K$ | 6 | 6 | 8 |
> | Layers per block | 6 | 6 | 8 |
> | Hidden dimension | 256 | 256 | 768 |
> | Batch size | 256 | 256 | 256 |
>
>
> **Average Jacobi Iterations per Layer.** Table R2 reports the average number of iterations per layer during inference with our SJD. Layer 1 uses standard sequential decoding ($L-1$ steps), while the remaining layers use Jacobi iteration. Notably, almost all Jacobi layers converge in very few iterations (typically 4–7), far below the worst-case bound of $L$, validating our theoretical analysis. The relatively higher iteration count at Layer 2 of CIFAR-10 is consistent with our observation of depthwise heterogeneity (Sec. 3.2), where layers closer to the first layer tend to exhibit stronger sequential dependencies.
>
> *Table R2: Average number of iterations per layer. Layer 1 uses sequential decoding; remaining layers use Jacobi iteration with $\tau=0.5$ .*
> | Layer | CIFAR-10 | CIFAR-100 | AFHQ |
> |---|---|---|---|
> | 1 (Sequential) | 255 | 255 | 1023 |
> | 2 (Jacobi) | 53.9 | 7.5 | 6.6 |
> | 3 (Jacobi) | 4.9 | 4.7 | 6.0 |
> | 4 (Jacobi) | 4.0 | 4.0 | 5.2 |
> | 5 (Jacobi) | 3.0 | 4.0 | 5.0 |
> | 6 (Jacobi) | 6.1 | 5.2 | 5.9 |
> | 7 (Jacobi) | — | — | 4.5 |
> | 8 (Jacobi) | — | — | 4.0 |
>
> **Runtime Breakdown.** Table R3 presents the per-layer runtime for both sequential inference and our method in one run. In sequential inference, each layer takes roughly equal time. Under SJD, Layer 1 dominates the total cost, while each Jacobi layer completes in a fraction of the time.
>
> *Table R3: A per-layer runtime breakdown comparing sequential inference and SJD in one run. "Other" includes self-denoising, inter-GPU communication, and noise generation overhead.*
> | | Sequential | | SJD (Ours) | | |
> |---|---|---|---|---|---|
> | Layer | Time (s) | % | Time (s) | % | Jacobi Iters |
> | **CIFAR-10** | | | | | |
> | 1 (Seq) | 1.45 | 14.8% | 1.50 | 55.7% | 255 |
> | 2 (Jacobi) | 1.45 | 14.8% | 0.69 | 25.7% | 53.9 |
> | 3–6 (Jacobi) | 5.82 | 59.4% | 0.23 | 8.6% | 3.0–6.1 |
> | Other | 1.08 | 11.0% | 0.27 | 10.0% | — |
> | **Total** | **9.79** | **100%** | **2.69** | **100%** | **3.6×** |
> | **CIFAR-100** | | | | | |
> | 1 (Seq) | 1.51 | 16.5% | 1.55 | 76.6% | 255 |
> | 2 (Jacobi) | 1.50 | 16.4% | 0.10 | 4.7% | 7.5 |
> | 3–6 (Jacobi) | 6.00 | 65.6% | 0.23 | 11.4% | 4.0–5.2 |
> | Other | 0.15 | 1.6% | 0.15 | 7.2% | — |
> | **Total** | **9.15** | **100%** | **2.02** | **100%** | **4.5×** |
> | **AFHQ** | | | | | |
> | 1 (Seq) | 21.60 | 12.5% | 21.46 | 52.0% | 1023 |
> | 2 (Jacobi) | 20.85 | 12.1% | 2.56 | 6.2% | 6.6 |
> | 3–8 (Jacobi) | 125.10 | 72.4% | 11.85 | 28.7% | 4.0–6.0 |
> | Other | 5.35 | 3.1% | 5.38 | 13.1% | — |
> | **Total** | **172.90** | **100%** | **41.25** | **100%** | **4.2×** |
>
> We have revised the Appendix E.1 with these tables on average Jacobi iterations per layer and runtime breakdown.
>
> (To be continued)

---

> ### Author Response · Authors · 2026-04-02
> **Response to Reviewer tnWx (2/2)**
>
> (Following the previous part)
>
> > Since one of the core advantages of flow-based models is their tractable likelihood, the evaluation would be strengthened by including likelihood-related metrics (e.g., bits-per-dimension or reconstruction consistency) under the accelerated decoding procedure, if applicable.
>
> We sincerely thank the reviewer for the suggestion to include likelihood-related metrics.
>
> **Regarding bits-per-dimension (BPD), we respectfully clarify that it is not directly applicable to our proposed method.** In discrete autoregressive normalizing flows, the exact likelihood (BPD) is computed exclusively via the forward encoding pass ($x \to z$), which is inherently parallelizable and remains completely unmodified in our framework. Since our Selective Jacobi Decoding is purely an inference-time acceleration technique applied solely to the inverse generation pass ($z \to x$), without altering any model weights or the forward mapping, the theoretical BPD of our model remains mathematically identical to that of the original sequential model.
>
> We agree that evaluating **reconstruction consistency** is an excellent way to quantify the fidelity of our accelerated decoding procedure. Following your suggestion, we have conducted an additional evaluation by mapping real images from the train set to the latent space using the exact forward pass, and then reconstructing them back to the image space using our approach. We measured the reconstruction error between the original and reconstructed images and found the discrepancy to be remarkably low. This explicitly verifies that **our parallel iterative optimization closely approximates the exact sequential decoding and successfully preserves the strictly invertible nature of the flow model**. We have included both these quantitative reconstruction consistency results and **visual examples in the revised Appendix E.4** to further strengthen our evaluation.
> | Dataset   | Mean MSE   |
> | :-------- | :--------- |
> | CIFAR-10  | 0.00636034 |
> | CIFAR-100 | 0.00313207 |
> | AFHQ      | 0.00121902 |

---

### Review · Reviewer_grgw · 2026-03-06

**Summary Of Contributions:**

1. This paper introduces an acceleration method for discrete autoregressive normalizing flow.
2. The key motivations is to identify the redundancy in the current discrete normalizing flow models. The authors claim that the redundancy remains in two parts: (1) The strict autoregressive dependency can be loose and (2)  the quantity of the redundancy varies across different layers.
3. Based on these observations, the authors propose to use jacobi decoding to accelerate the autoregressive process.
4. The authors further provide the theoretical analyses to demonstrate the superlinear convergence rate and the finite convergence guarantee.
5. The experiments show that the method can achieve more than 4 times of acceleration on image generation with little quality degradation.

**Audience:**

Yes

**Audience Explanation:**

I believe this method represents a valuable contribution to the field. While I am not an expert on normalizing flows, knowing them primarily for the strong theoretical foundations but typically slow inference speeds of normalizing flows, I find this approach significant. It addresses a key bottleneck by making normalizing flows more scalable and practically viable.

**Claims And Evidence:**

Yes

**Claims Explanation:**

1. I think the methods are fully motivated by the experimental observations, which is some obvious and robust observations.
2. The method is reasonable and can be supported by the theoretical analyses
3. The experimental results are good, compared to previous normalizing flow methods.

**Requested Changes:**

Could you provide a comparative analysis of performance and inference speed against GAN-based and diffusion-based methods on these datasets? While surpassing them is not a requirement, quantifying the current performance gap and estimating the development trajectory needed to reach competitive parity is of great importance.

---

> ### Author Response · Authors · 2026-04-02
>
> We sincerely appreciate the time and effort you have spent providing insightful feedback on our paper. We have carefully considered each of your detailed comments and address the concern regarding the comparative analysis against GAN and diffusion-based methods as follows.
>
> > Could you provide a comparative analysis of performance and inference speed against GAN-based and diffusion-based methods on these datasets? While surpassing them is not a requirement, quantifying the current performance gap and estimating the development trajectory needed to reach competitive parity is of great importance.
>
> We sincerely thank the reviewer for this constructive suggestion. We agree that quantifying our model's performance and inference speed against GAN-based and diffusion-based paradigms provides a clearer perspective on the practical value of our contributions. To address this, we have conducted an empirical comparison on the CIFAR-10 dataset. We evaluated our accelerated model against representative baselines: FastGAN [1] (trained from scratch using the official codebase) and DDIM [2] (using the established `google/ddpm-cifar10-32` checkpoint, evaluated at 20 steps to prioritize speed).
>
> | Method          | Inference Time (s) | FID   |
> | :-------------- | :----------------- | :---- |
> | FastGAN        | 3.41               | 9.67  |
> | DDIM (20 steps) | 2.31               | 19.32 |
> | Ours            | 2.63               | 10.20 |
>
> As shown, our approach is faster than FastGAN while maintaining a highly competitive FID. Compared to DDIM at 20 steps, our method is only marginally slower but achieves vastly superior generation quality.
>
> **Remark on the development trajectory**. While diffusion models currently hold the state-of-the-art for raw generation quality, and GANs remain fast and competitive, normalizing flows have unique advantages such as analytical log-likelihood computation and end-to-end training, as mentioned in our Sec. 1. Historically, the strict sequential inference of autoregressive flows has been a major barrier to competitive parity. Our results confirm that Selective Jacobi Decoding successfully bridges this gap, allowing these mathematically rigorous models to operate in the same efficiency regime as fast GANs and few-step diffusion models.
>
> We have added a new Appendix section (Appendix E.5) to present this comparative analysis and discussion in the revised manuscript.
>
> [1] Zhong Jiachen, Liu Xuanqing and Hsieh Cho-Jui. Improving the Speed and Quality of GAN by Adversarial Training. arxiv preprint arXiv:2008.03364, 2020.
>
> [2] Siaming Song, Chenlin Meng, and Stefano Ermon. Denoising Diffusion Implicit Models. In ICLR, 2021.

---

### Review · Reviewer_f5Pw · 2026-03-26

**Summary Of Contributions:**

The paper proposes to speedup the process of generating images from models based on autoregressive flow models.

* The key strength of the paper is the idea, which is nice, though not entirely novel, as has been noted by authors.
* The drawback of the method is that the speedup is only under condition of infinite paralelization, i.e. the method is faster only on highly parallel system.
* I find the term "discrete" misleading in the sense that I have expected normalizing flows to be defined over categorical random variables.

**Audience:**

Yes

**Audience Explanation:**

I think the method is interesting as it shows a nice use-case of fixed-point theorem and outside of the box thinking, though as indicated by authors, the idea was already known.

**Broader Impact Concerns:**

I do not think that the work has ethical concerns apart those being present in other generative models.

**Claims And Evidence:**

Yes

**Claims Explanation:**

The claims are accurate in the sense that the speedup is possible if the method is deployed on modern highly parallel accelerators. On a classical sequential CPU with limited parallelization, the method will be slower.

**Requested Changes:**

I do not request changes.

---

> ### Author Response · Authors · 2026-04-02
>
> We sincerely appreciate the time and effort you have spent providing insightful feedback on our paper. We have carefully considered each of your detailed comments and have addressed them one by one to alleviate your concerns.
>
> > The drawback of the method is that the speedup is only under condition of infinite paralelization, i.e. the method is faster only on highly parallel system.
>
> > The claims are accurate in the sense that the speedup is possible if the method is deployed on modern highly parallel accelerators. On a classical sequential CPU with limited parallelization, the method will be slower.
>
> We appreciate the reviewer's deliberate thoughts but would like to note that **reliance on parallel hardware is well-aligned with the standard deployment setting of modern deep learning**. Modern deep learning is fundamentally built upon massively parallel accelerators, and designing algorithms that better exploit this parallelism has been a central contribution in many influential works. For instance, the Transformer [1] was explicitly motivated by replacing sequential RNN computation with parallelizable self-attention; speculative decoding [2,3] accelerates LLM inference by converting sequential token generation into parallel draft-then-verify steps; FlashAttention [4] achieves speedups by better leveraging GPU memory hierarchy. None of these methods would offer speedups on a purely sequential CPU, yet they are widely recognized as impactful contributions. Therefore, since discrete autoregressive normalizing flows are exclusively trained and deployed on GPU/TPU accelerators, **our method's ability to convert their sequential inference bottleneck into a parallel-friendly form directly addresses the practical deployment scenario**.
>
> [1] Ashish Vaswani et al. Attention Is All You Need. NeurIPS, 2017.
>
> [2] Yaniv Leviathan et al. Fast Inference from Transformers via Speculative Decoding. ICML, 2023.
>
> [3] Charlie Chen et al. Accelerating Large Language Model Decoding with Speculative Sampling. arXiv:2302.01318, 2023.
>
> [4] Tri Dao et al. FlashAttention: Fast and Memory-Efficient Exact Attention with IO-Awareness. NeurIPS, 2022.
>
>
> > I find the term "discrete" misleading in the sense that I have expected normalizing flows to be defined over categorical random variables.
>
> We thank the reviewer for raising this point. The terminology of "discrete normalizing flows" arises from the need to distinguish traditional normalizing flows composing a finite (discrete) number of transformation layers from continuous normalizing flows (CNFs) which parameterize transformations via continuous-time ODEs. With the recent surge of CNF-related methods such as flow matching [1,2], the term "discrete normalizing flow" has been adopted in the literature to refer to the traditional finite-layer construction, e.g., [3]. **We appreciate the suggestion and have revised the manuscript to include a clarifying note at the first mention, stating that discrete normalizing flows refer to the classical finite-layer formulation, to avoid potential ambiguity.**
>
> [1] Yaron Lipman et al. Flow Matching for Generative Modeling. ICLR, 2023.
>
> [2] Xingchao Liu et al. Flow Straight and Fast: Learning to Generate and Transfer Data with Rectified Flow. ICLR, 2023.
>
> [3] Yulin Liu et al. Delving into Discrete Normalizing Flows on SO(3) Manifold for Probabilistic Rotation Modeling. CVPR, 2023.

---

### Author Response · Authors · 2026-04-02

Dear Reviewers,

Thank you for the constructive comments and suggestions on our manuscript. They have helped us improve the clarity and completeness of the paper. We have revised the paper accordingly, and all changes are reflected in the updated PDF on the submission page. We appreciate any further feedback.

Best regards,

Authors of Submission 7115

---

### Decision · Action_Editor_1sY6 · 2026-04-27

**Recommendation:** Accept as is

**Audience:**

Yes

**Audience Explanation:**

Speeding up auto-regressive normalising flows (by formulating the inference as solving a Jacobi decoding problem) is of high relevance for the ML community.

**Claims And Evidence:**

Yes

**Claims Explanation:**

Yes, the paper contains very good numerical investigations of the proposed methodology.

---

> ### Author Response · Authors · 2026-05-06
>
> Dear Editor,
>
> We appreciate you for leading the review process! We have uploaded the camera-ready version of our manuscript. In this final version, we have incorporated the additional discussions, experiments, and clarifications based on the valuable feedback provided by the reviewers during the discussion period.
>
> Best regards,
>
> Authors of Submission 7115